# Study on the Influence of Host–Guest Structure and Polymer Introduction on the Afterglow Properties of Doped Crystals

**DOI:** 10.3390/molecules29194537

**Published:** 2024-09-24

**Authors:** Wenhui Feng, Zongyong Lou, Xiaoqiang Zhao, Mingming Zhao, Yaqin Xu, Yide Gao

**Affiliations:** 1Department of Thermal Engineering, Hebei Petroleum University of Technology, Chengde 067000, China; 2School of Chemical Engineering and Technology, Tianjin University, Tianjin 300354, China

**Keywords:** long-persistent luminescence materials, organic small-molecule room-temperature phosphorescence, donor–acceptor charge separation, polymer

## Abstract

Due to their low cost, good biocompatibility, and ease of structural modification, organic long-persistent luminescence (LPL) materials have garnered significant attention in organic light-emitting diodes, biological imaging, information encryption, and chemical sensing. Efficient charge separation and carrier migration by the host–guest structure or using polymers and crystal to build rigid environments are effective ways of preparing high-performance materials with long-lasting afterglow. In this study, four types of crystalline materials (MODPA: DDF-O, MODPA: DDF-CHO, MODPA: DDF-Br, and MODPA: DDF-TRC) were prepared by a convenient host–guest doping method at room temperature under ambient conditions, i.e., in the presence of oxygen. The first three types exhibited long-lived charge-separated (CS) states and achieved visible LPL emissions with durations over 7, 4, and 2 s, respectively. More surprisingly, for the DDF-O material prepared with PMMA as the polymer substrate, the afterglow time of DDF-O: PMMA was longer than 10 s. The persistent room-temperature phosphorescence effect caused by different CS state generation efficiencies and rigid environment were the main reason for the difference in LPL duration. The fourth crystalline material was without charge separation and exhibited no LPL because it was not a D-A system. The research results indicate that the CS state generation efficiency and a rigid environment are the key factors affecting the LPL properties. This work provides new understandings in designing organic LPL materials.

## 1. Introduction

Organic long-persistent luminescence (LPL) materials are being increasingly researched and developed due to their sustainable preparation process and easily controllable luminescence [1,2,3,4]. These materials can exhibit LPL under ambient conditions after photo-excitation. Compared with inorganic long-afterglow materials, organic LPL materials offer the advantages of a conveniently adjustable emission lifetime and wavelength, and stimulus-responsive emission characteristics [5,6]. These features provide unique benefits in applications such as information encryption, anti-counterfeiting, biosensing, and bioimaging [7,8,9,10]. Early studies on organic long-afterglow materials are mainly based on metal complexes such as Ir and Pt. Since heavy metals can promote spin-orbit coupling, singlet excitons can easily be converted into triplet excitons through intersystem crossing (ISC), and the rate of triplet excitons radiating back to the ground state is fast [11,12]. However, the cost of these heavy metals is relatively high. In recent years, pure organic materials have become a new research focus [9,10,11,12]. However, organic small molecules are affected by spin-forbidden transitions and weak spin-orbit coupling constants, which lowers the rate of tampering between the S1-T1 systems and makes it difficult to achieve a long afterglow time.

Researchers have developed various methods to obtain highly efficient organic phosphorescent materials with ultra-long afterglow. The persistent phosphorescence effect of pure organic compounds can be induced by accurately controlling the ISC constant and suppressing non-radiative decay [12,13,14]. Examples of these compounds include small-molecule crystals, metal–organic frameworks, hydrogen-bonded organic frameworks, ionic crystals, carbon dots, cocrystals, host–guest doped materials, and polymer matrices [15,16,17,18,19,20]. These configurations can stabilize triplet excitons under different environments and achieve high-performance persistent phosphorescence. Among the reported strategies, organic small-molecule crystals show great potential in achieving efficient LPL [21,22,23,24], which can function in rigid environments, inhibit non-radiative transitions, reduce the influences of external environments, and prevent the quenching of triplet excitons by water and oxygen. However, it is still difficult for pure organic small-molecule crystals to achieve the afterglow effect.

Host–guest doping promotes electron charge transfer and facilitates the formation of long-life charge separation states, or non-radiative transitions are inhibited by a rigid environment constructed by a polymer matrix, so there are effective methods for constructing long-life afterglow materials [25,26,27]. However, triplet excitons are easily quenched by oxygen or other non-radiative deactivation pathways, indicating that the luminescence properties of pure organic phosphorescent materials still need improvement. Additionally, many studies have explored host–guest organic long-afterglow crystal materials, but how the molecular structure affects the afterglow and the potential mechanisms is unclear. Discussions on how the structure of the guest material and its properties (e.g., energy level, carrier mobility, and stability) affect the afterglow properties of doped crystals are rare. In particular, the theoretical system of changing the afterglow time by adjusting the material structure still needs to be developed.

By adopting a general host–guest doping strategy and changing the molecular structure, we construct a time-adjustable and efficient room-temperature phosphorescence (RTP) system. Fluorene molecules modified by different functional groups (Figure 1a) are doped into a diphenylamine host which is easy to crystallize. Effective host–object doping is realized with simple water-assisted crystallization, which induces the long afterglow performance of organic doped crystals under ambient conditions and produces an RTP system with fine, adjustable optical properties. It can be seen from Figure 1b that three kinds of crystals, namely, MODPA: DDF-O, MODPA: DDF-CHO, and MODPA: DDF-Br, exhibit a green afterglow of up to 7, 4, and 2 s, respectively. In addition, MODPA: DDF-TRC exhibits a green afterglow shorter than 0.1 s after being excited by a 365 nm ultraviolet (UV) lamp, which is also confirmed by the decay spectrum of the afterglow performance from the long-afterglow material (Figure 1c). This is similar to the afterglow time observed by the naked eye, demonstrating that the change in the guest material leads to a significant difference in LPL properties. What is more surprising is that the residual glow time of the materials prepared by constructing a rigid environment with a polymer matrix is more than 10 s, which further proves that a rigid environment is conducive to inhibiting non-radiative transition. To highlight the application advantages of the crystals with an adjustable afterglow time, we prepared anti-counterfeiting patterns with dynamic afterglow time gradients.

## 2. Results and Discussion

To better understand the application potential of fluorene-based small-molecule series crystals with long afterglow, it is crucial to comprehensively examine their properties and understand how these properties are influenced by various structural factors. First, the structure–activity relationship of the fluorene-based small-molecule crystals must be studied. This includes clarifying how structural changes in the host–guest compounds and crystal configuration changes in the donor–acceptor (D-A)-type doped crystals affect LPL performance. Also, identifying corresponding control methods is key to the practical application of this series of compounds.

Thermodynamic stability is critical to the performance of materials. The TGA and DSC properties of fluorene-based small molecules DDF-O, DDF-CHO, DDF-Br, and DDF-TRC were tested at a heating rate of 10 °C·min^−1^ in a nitrogen atmosphere. From the TGA curves in Figure 2a, it can be seen that the temperatures corresponding to 5% mass loss of DDF-O, DDF-CHO, DDF-Br, and DDF-TRC compounds are 350, 367, 372, and 348 °C, respectively. The DSC curves in Figure 2b show that the melting points of DDF-O, DDF-CHO, DDF-Br, and DDF-TRC are 185, 235, 288, and 220 °C, respectively, indicating good thermal stability for the fluorene-based small molecules. The FL microscopy results show that the doped crystals have a sheet structure (Figure 2c and Appendix A). Charge carrier mobility is another factor to be considered to achieve satisfactory performance of host and guest-type afterglow materials. Therefore, we evaluated the carrier transport performance of the guest material that was formed by spin-coating and sandwiching between two electrodes, as well as spin-coating to form hole-dominated devices (ITO/PSS:PEDOT/guest/Au). The steady-state SCLC technique was employed to measure mobilities at room temperature. Interestingly, the hole mobilities of DDF-O, DDF-CHO, DDF-Br, and DDF-TRC were 7.5 × 10^−4^, 2.9 × 10^−4^, 1.3 × 10^−4^, and 5.4 × 10^−5^ cm^2^·V^−1^·s^−1^, respectively, which are considered as moderate values. The hole carrier mobility of the four compounds was gradually weakened (DDF-O > DDF-CHO > DDF-Br > DDF-TRC).

The UV–Vis absorption spectra of all monomer molecules were tested (Appendix A). Additionally, the UV–Vis absorption, FL, and phosphorescence spectra of the doped crystals MODPA: DDF-O, MODPA: DDF-CHO, MODPA: DDF-Br, and MODPA: DDF-TRC were tested (Figure 3a). The UV–Vis absorption spectrum characteristics of all monomer molecules are shown in Appendix A. The absorption peaks of MODPA, DDF-O, DDF-CHO, DDF-Br, and DDF-TRC monomers in toluene are located at 299, 368, 383, 373, and 389 nm, respectively, as affected by the short π connections of the compounds [22]. Compared to DDF-O, the absorption peaks of the DDF-CHO, DDF-Br, and DDF-TRC compounds are considerably broadened and red-shifted by up to 21 nm, which is attributed to the greater number of aldehyde groups and bromine atoms, and 4, 6-triazine withdrawing capacity of DDF-CHO, DDF-Br, and DDF-TRC, respectively [27]. In the crystalline state (Figure 3a), the absorption peaks are red-shifted by 19, 22, 27, and 80 nm for DDF-O, DDF-CHO, DDF-Br, and DDF-TRC, respectively, compared to their toluene solutions. This shift is due to the strong intralayer π–π interactions in the crystalline state [27], which implies that the absorption peaks at 387 nm for MODPA:DDF-O, at 405 nm for MODPA:DDF-CHO, at 400 nm for MODPA:DDF-Br, and at 469 nm for MODPA:DDF-TRC can be attributed to the DDF-O, DDF-CHO, DDF-Br, and DDF-TRC modules, respectively [27].

To obtain the HOMO energy level (E_HOMO_) and LUMO energy level (E_LUMO_) of the monomer materials, we used the BAS 100 W electrochemical analyzer to test the CV curves of DDF-O, DDF-Br, DDF-CHO, DDFTRC, and MODPA monomers (Appendix A and Table 1). All the monomer molecules display good oxidation peaks. Through simple analysis, the HOMO energy levels of DDF-O, DDF-Br, DDF-CHO, DDFTRC, and MODPA were obtained as −5.10, −5.20, −5.30, −5.33, and −5.70 eV, respectively. Using the formula E_g_ = E_HOMO_ − E_LUMO_, the LUMO energy levels of DDF-O, DDF-Br, DDF-CHO, DDF-TRC, and MODPA were calculated as −1.88, −2.14, −2.15, −3.79, and −2.15 eV, respectively. As shown in Table 1, DDF-O, DDF-CHO, and DDF-Br have higher LUMO energy levels than MODPA. Therefore, DDF-O, DDF-CHO, and DDF-Br can act as donors, while MODPA can serve as the acceptor, allowing their energy levels to match and form D-A systems. However, the LUMO energy level of DDF-TRC is lower than that of MODPA, so DDF-TRC and MODPA cannot form a D-A system. This discrepancy is likely the main reason why the MODPA: DDF-TRC system fails to exhibit afterglow performance [24].

The FL emission attenuation curve and phosphorescence emission decay spectra further prove that a charge-separated (CS) state occurs between host and guest. As shown in Figure 3b, the FL emission of the DDF-O module at 420 nm is double exponentially attenuated, and its emission lifetime consists of a major, fast component of 1.34 ns (97.0%) and a small, slow component of 3.72 ns (3.0%). The lifetime of 1.34 ns is the result of the transfer of light-induced electrons from the excited DDF-O to the MODPA module that forms a charge-separated (CS) state, while the lifetime of 3.72 ns is due to the intersystem transition from singlet to triplet, so the RTP emission of this system originates from the interaction between the MODPA and DDF-O modules when the doped crystals are excited at 365 nm. This interaction in rigid crystals is attributed to the charge separation behavior given to the acceptor. The major and fast component of fluorescence emission lifetimes, 1.34 ns (97.0%) for MODPA: DDF-O, 1.27 ns (93.0%) for MODPA: DDF-CHO and 1.12 ns (90.3%) for MODPA: DDF-Br, were ascribed to the photo-induced electron transfer from the excited DDF-O (DDF-CHO and DDF-Br) to the MODPA module to form the CS states. The minor and slow component of fluorescence emission lifetimes, 3.72 ns (3.0%) for MDPA: DDF-O, 3.38 ns (7.0%) for MDPA: DDF-CHO and 3.65 ns (9.7%) for MDPA: DDF-Br, were attributed to the singlet–triplet conversion [28,31,32]. This indicated that the charge separation efficiency of the MDPA: DDF-O crystal (97.0%) was better than that of the MDPA: DDF-CHO crystal (93.0%) and MDPA: DDF-Br crystal (90.3%) [24].

Phosphorescence emission decay spectra exhibit bi-exponential decay or single-exponential decay of the RTP emission at 520 nm with lifetimes of 433.22 ms (97.9%) and 47.30 ms (2.1%) for MODPA:DDF-O, 330.62 ms (97.0%) and 23.80 ms (3.0%) for MODPA: DDF-CHO, 167.25 ms (97.0%) and 13.20 ms (3.0%) for MODPA: DDF-Br, and 79.69 ms (100%) for MODPA: DDF-TRC (Figure 3c). The significant RTP emission of three kinds of excited doped crystals includes that of (1) the MODPA triplet states and (2) that of the DDF-O (DDF-CHO, DDF-Br) triplet states, produced by the CS states (Figure 3c), and the difference in phosphorescence lifetime could affect the LPL duration time. Here, we proposed a long-afterglow luminescence mechanism of the doped crystal MODPA: DDF-O, as shown in Figure 3d. Since the energy of the CS state is higher than that of the MODPA triplet state and the DDF-O triplet state, the conversion from the former state to the latter states is allowed [24].

The CS states are directly related to the LPL performance. The MODPA: DDF-TRC crystal is not a D-A system, for which photoinduced electron or hole transfer is forbidden at 365 nm excitation, so the CS states do not appear in MODPA: DDF-TRC. The semi-logarithmic plot of the LPL emission in Figure 1d does not follow an exponential decay profile but shows that the LPL of MODPA: DDF-O and MODPA: DDF-CHO can continue for more than 7 s and 4 s, respectively [1,16]. This is dissimilar to the RTP emission (Figure 3c) and further indicates the existence of the CS states with weak delayed fluorescence [22]. Although the doping ratios and preparation methods of the MODPA:DDF-O, MODPA:DDF-CHO, and MODPA:DDF-Br crystals are consistent, the structural differences between DDF-O, DDF-CHO, and DDF-Br still lead to significant variations in their LPL performance. This should be related to the generation efficiency of long-lived CS states by variation in DDF derivatives.

To better reflect the advantages of the fluorine-based materials and their prospects in commercial applications, we used DDF-O as the guest and polymer poly (methyl methacrylate (PMMA) as the host to prepare rigid (MODPA: DDF-O: PMMA). Surprisingly, when the polymer substrate is constructed in a rigid environment to prepare the afterglow material, the afterglow time is further enhanced (Figure 4a); the afterglow time of MODPA: DDF-O: PMMA is longer than 10 s, which indicates that this kind of PMMA material has a good ability to construct a rigid environment. This is mainly because the construction density of the rigid environment based on the polymer is better, which can just inhibit the non-radiative transition of the trilinear molecules, which further increases the phosphor lifetime from 433 ms (MODPA: DDF-O) (Figure 3c) to 836 ms (MODPA: DDF-O: PMMA) (Figure 4c), which is consistent with its afterglow performance. Compared with the crystal material MODPA: DDF-CHO, MODPA: DDF-CHO: PMMA also has enhanced afterglow performance. Figure 4b shows the semi-logarithmic plots of the emission decay profiles of MODPA: DDF-O: PMMA and MODPA:DDF-CHO: PMMA from −2 s to 10 s. Both MODPA: DDF-O: PMMA and MODPA: DDF-CHO: PMMA exhibit photoluminescent characteristics under light excitation (−2 to 0 s), and long-afterglow characteristics after turning off the excitation light source (0–12 s). The long-persistent luminescence emissions of MODPA: DDF-O: PMMA and MODPA: DDF-CHO: PMMA visibly endured for approximately 10 s and 6 s, respectively, consistent with the afterglow times under naked eye observation. This result shows that, although the rigid environment can improve the afterglow properties of the materials, it is still closely related to the charge transfer of the host and guest materials. The material prepared on the PMMA substrate is a rigid film suitable for applications requiring greater hardness. We also prepared polymer films on substrates of polyvinyl alcohol (PVA) (MODPA: DDF-O: PVA) and polyethylene glycol (PEG) (MODPA: DDF-O: PEG) (Figure 4a). The films were simply and quickly prepared by heating the compounds. As expected, the PVA substrate realized a flexible film with similar afterglow times as PMMA (8 s for MODPA: DDF-CHO: PVA; see upper row of Figure 4a).

We combined the four crystals with different afterglow durations, using different combinations (Figure 5) to highlight the special advantages of dynamic time in anti-counterfeiting and information encryption applications. As shown in Figure 5a, MODPA: DDF-O, MODPA: DDF-CHO, MODPA: DDF-Br and MODPA: DDF-TRC designed and produced the “2023” digits under the excitation of a 365 nm ultraviolet lamp. These signature locks were activated once the UV lamp excitation stopped. MODPA: DDF-O formed the 2 lock, MODPA: DDF-CHO formed the 0 lock, MODPA: DDF-Br formed the 2 lock, and MODPA: DDF-TRC formed the 3 lock due to its continuously decreasing afterglow life. Only the 2 lock is effective because it is the longest-lived RTP emission; therefore, we observed the final message 2, which improves the security of the encrypted information. At the same time, in order to better demonstrate the application scenarios of this kind of afterglow material, we mixed the material with PVA to make a paste, which was coated on different media and could be written on paper and mosaic leaves to prepare encrypted files. It can also be coated on seabed shells, showing its potential for application in water, as well as for the preparation of flexible and rigid films.

## 3. Experiments

### 3.1. Materials

MODPA was purchased from Energy Chemical (Shanghai, China) and purified by recrystallization. DDF-O, DDF-CHO, and DDF-Br were purchased from Alpha (Zhengzhou, China), whereas DDF-TRC was synthesized using reagent-grade compounds and solvents according to Refs. [33,34,35], and these compounds were further purified by recrystallization. Polyethylene pyrrolidone (PVP, average molecular weight: 1,300,000) was purchased from Sinopharm Chemical Reagent Co., Ltd. (Shanghai Province, country China). PMMA (average molecular weight: 130,000) was purchased from Tianjin Heowns Biochemical Technology Co., Ltd. (Tianjin, China). PEG 3000 (average molecular weight: 3000) was purchased from Tianjin Heowns Biochemical Technology Co., Ltd. Poly (vinyl alcohol) (average molecular weight: 2000; degree of hydrolysis: 87–89%) was purchased from Shanghai meryer Co., Ltd. (Shanghai, China)

Preparation of trace-doped crystal MODPA: DDF-O. At room temperature, 5 mL of anhydrous ethanol was added to 99 mol% MODPA:1%DDF-O mixed powder (0.50 g), and the solution was prepared using an ultrasonic cleaner to dissolve and uniformly mix the donor and acceptor materials. Then, 50 mL of deionized water was slowly added to the solution in the presence of air. The mixture was then stirred continuously at 95 °C for 20 min. Afterwards, the system was cooled to room temperature with stirring. The suspension was left at room temperature for 6 h to allow the sediment to settle completely. The resulting solid was collected by filtration, yielding a white crystalline product.

Preparation of trace-doped crystals MODPA: DDF-CHO, MODPA: DDF-Br, and MODPA: DDF-TRC. The preparation process of MODPA: DDF-CHO, MODPA: DDF-Br, and MODPA: DDF-TRC was similar to that of MODPA: DDF-O, which only replaced the DDF-O in the raw material with DDF-CHO, DDF-Br, and DDF-TRC, respectively.

Preparation of trace-doped crystal DDF-O: PMMA. DDF-O, MODPA, and PMMA were measured at a mass ratio of 1:30:100 and added to a 10 mL vial. The vial was then placed on a constant temperature heating table, heated and melted at 160 ℃ for 20 min, and cooled for later use.

Preparation of trace-doped crystal DDF-O: PEG. DDF-O, MODPA, and PEG were also measured at a mass ratio of 1:30:100 and added to a 10 mL vial. The vial was then placed on a constant temperature heating table, heated and melted at 180 °C for 15 min, and cooled for later use.

Preparation of trace-doped crystal DDF-O: PVA. DDF-O, MODPA, and PVA with a mass ratio of 1:30:100 were weighed and placed into a 10 mL vial. The vial was then set on a constant temperature heating table, heated and melted at 190 ℃ for 25 min, and cooled for later use.

Preparation of the MODPA: DDF-O: PVP Paint: 1.0 g of the trace-doped (5 mol‰) MODPA: DDF-O powder and 3.0 g of PVP were added into dichloromethane (8.0 mL); then the mixture was stirred at room temperature to produce the paint in which the mass percentage of host–guest doping material was 25 wt% (without solvent).

### 3.2. Measurement and Characterization

Thermogravimetric analysis (TGA) was conducted using a Rigaku Thermo Plus TGA 2 instrument (manufacturer Rigaku Corporation, Tokyo, Japan), and differential scanning calorimetry (DSC) was performed with a TA DSC Q20 instrument (manufacturer Rigaku Corporation, Tokyo, Japan). Electrochemical properties were measured using cyclic voltammetry (CV) on a BAS 100 W electrochemical analyzer (manufacturer Du Pu Instrument Co., Ltd., Zhengzhou, China). The room-temperature photoluminescence (PL), phosphorescence, and LPL spectra of the crystals were recorded with an Ocean Optics fiber spectrophotometer (manufacturer thermo Co., Ltd., Berlin, Germany). Ultraviolet–visible (UV–Vis) spectroscopy was performed using a Thermo Spectronic Helios Gamma spectrometer (manufacturer thermo Co., Ltd., Berlin, Germany). The quartz cells had a path length of 1 cm. Fluorescence (FL) spectroscopy was carried out using a Varian CARY ECLIPSE fluorescence spectrometer (manufacturer Agilent Technologies Inc., Palo Alto, CA, USA). Time-resolved FL and PL experiments were performed with a spectrophotometer (Gilden Photonics, London, UK) using a pulsed source at 480 nm (BDS-SM ps diode laser, Gilden Photonics, London, UK). The time-resolved signals were recorded by a time-correlated single-photon counting detection technique. SEM images were recorded via a Hitachi JSM-7800F field emission microscope equipped with an EDX spectrometer (manufacturer Japan Electronias Co., Ltd., Tokyo, Japan).

Space-charge limited current (SCLC): The hole-only devices were fabricated with the ITO/PEDOT:PSS/blend/Au configuration. The PEDOT:PSS buffer layer was prepared by spin-coating a 3.5 wt% PEDOT:PSS isopropanol solution onto a pre-cleaned ITO substrate which was then baked at 150 °C for 10 min. Subsequently, the blend was spin-coated under the same conditions used for preparing optimal solar cells. The Au layer was thermally deposited on the top of the blend in a vacuum. The Au layer was deposited at a low speed to avoid the penetration of Au atoms into the active layer. The current density-voltage curves of the devices were recorded with a Keithley 2400 source.

## 4. Conclusions

In this paper, four crystalline materials, MODPA: DDF-O, MODPA: DDF-CHO, MODPA: DDF-Br, and MODPA: DDF-TRC, were prepared by a convenient host–guest doping method at room temperature in the presence of oxygen. Only the first three crystals exhibited visible LPL emissions, with durations over 7, 4, and 2 s, respectively. More importantly, the afterglow time of MODPA: DDF-O: PMMA exceeded 10 s. The research results indicate that the CS state generation efficiency and a rigid environment are the key factors affecting the LPL properties. Efficient charge separation and carrier migration by the host–guest structure or using polymers and crystal to build rigid environments are effective ways of preparing high-performance materials with long-lasting afterglow. This study provides new understandings in designing organic LPL materials. At the same time, flexible and rigid films were developed. The afterglow time and brightness of green afterglow materials have been significantly enhanced; however, further advancements are required in the development of other color afterglow materials. Firstly, one of the common applications of organic RTP materials is biological imaging, while the beneficial emission wavelength is in the near-infrared region. However, there are only a few related studies at present that seek to identify a suitable host matrix as an important way to activate the near-infrared RTP emission. Secondly, blue is one of the three primary colors, and it is a crucial element in the regulation and application of organic room-temperature phosphorescence (ORTP). However, the considerable Stokes shift of small organic molecules presents a challenge for creating blue afterglow materials.

## Figures and Tables

**Figure 1 molecules-29-04537-f001:**
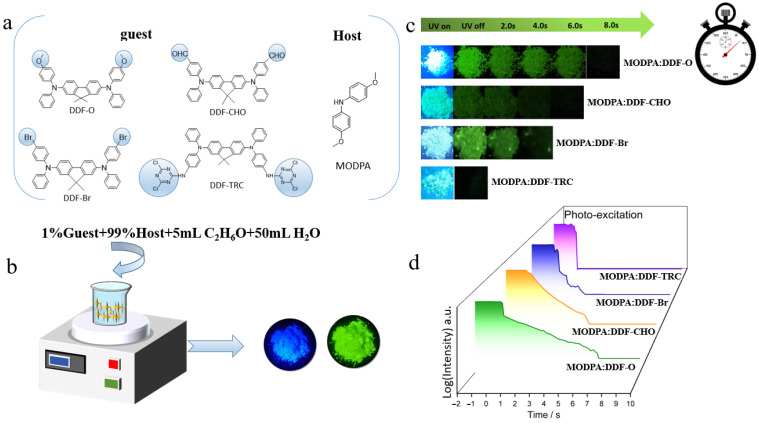
(**a**) Chemical structures of MODPA, DDF-O, DDF-CHO, DDF-Br and DDF-TRC. (**b**) The 1% DDF-doped MODPA crystals are prepared by a “green” water-based method. (**c**) The LPL photograph under ambient conditions of four doped crystals (excitation: 365 nm). (**d**) The semi-logarithmic plot of the emission decay profile of the four doped crystals from −2 s to 10 s, which exhibited photoluminescence (PL) upon photo-excitation (from −2 s to 0 s) and LPL after the excitation turned off (excitation wavelength: 365 nm; excitation power: 10 mW; excitation time: 2 s; sample temperature: 300 K).

**Figure 2 molecules-29-04537-f002:**
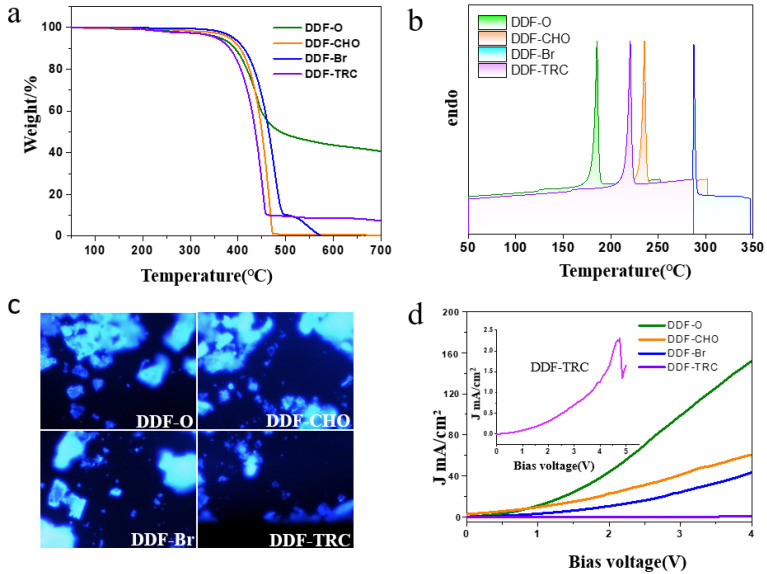
(**a**) TGA and (**b**) DSC curves of the compounds DDF-O, DDF-CHO, DDF-Br, and DDF-TRC. (**c**) Photographs of LPL from the MODPA: DDF-O, MODPA: DDF-CHO, MODPA: DDF-Br and MODPA: DDF-TRC doped crystals under a fluorescence microscope at 365 nm excitation. (**d**) The current density (J) versus voltage curves (V) (J-V) plots of the single-charge carrier for DDF-O, DDF-CHO, DDF-Br, and DDF-TRC by SCLC technique. (The illustration is a magnification of compound DDF-TRC).

**Figure 3 molecules-29-04537-f003:**
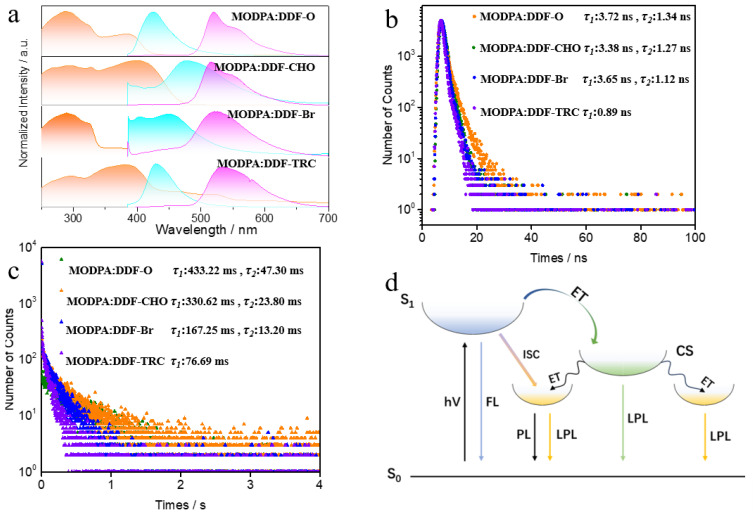
(**a**) UV–Vis absorption spectra (orange), fluorescence (cyan), and phosphorescence spectra (magenta) of the MODPA: DDF-O, MODPA: DDF-CHO, MODPA: DDF-Br and MODPA: DDF-TRC doped crystals (excitation: 365 nm), measured at room temperature under ambient conditions. (**b**) Fluorescence emission decay spectra of the MODPA: DDF-O, MODPA: DDF-CHO, MODPA: DDF-Br, and MODPA: DDF-TRC doped crystals (excitation: 365 nm, emission: 420 nm). (**c**) Phosphorescence emission decay spectra of the MODPA: DDF-O, MODPA: DDF-CHO, MODPA: DDF-Br, and MODPA: DDF-TRC doped-crystals (excitation: 365 nm, emission: 520 nm). (**d**) Energy diagram and photophysical processes of the MODPA: DDF-O-doped crystals (excitation: 365 nm).

**Figure 4 molecules-29-04537-f004:**
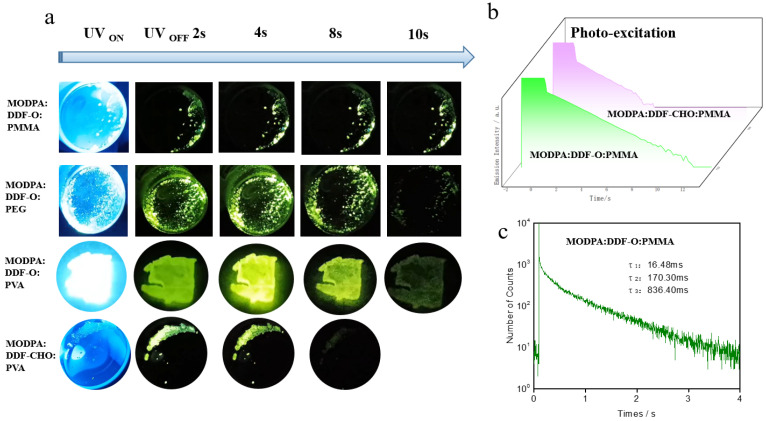
(**a**) The rigid and flexible thin-film afterglow materials were prepared by using PMMA, PEG and PVA polymers as hosts and DDF-O or DDF-O as guest dopants. (**b**) The semi-logarithmic plot of the emission decay profile of the four doped crystals from −2 s to 10 s, which exhibited photoluminescence (PL) upon photo-excitation (from −2 s to 0 s) and LPL after the excitation turned off (excitation wavelength: 365 nm; excitation power: 10 mW; excitation time: 2 s; sample temperature: 300 K). (**c**) Phosphorescence emission decay spectra of the MODPA: DDF-O: PMMA, MODPA: DDF-CHO: PMMA-doped materials (excitation: 365 nm, emission: 520 nm).

**Figure 5 molecules-29-04537-f005:**
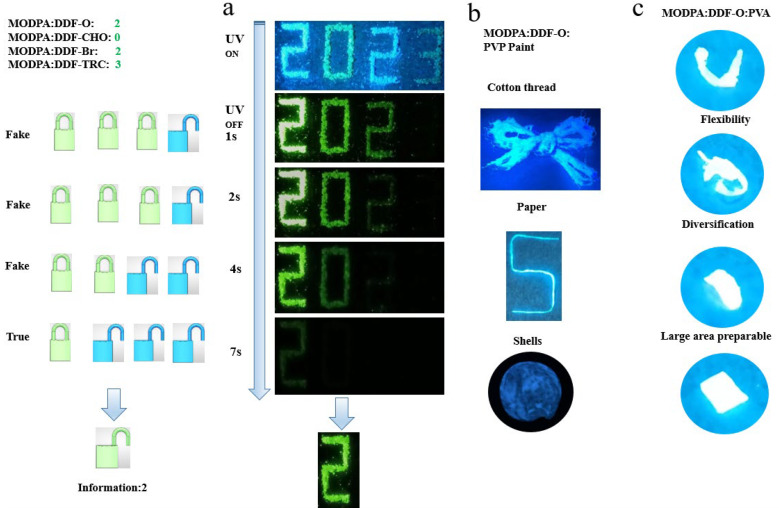
(**a**) Illustration of multiple information encryption of “2023” digits, based on the doped crystals of MODPA: DDF-O, MODPA: DDF-CHO, MODPA: DDF-Br, and MODPA: DDF-TRC. (**b**) MODPA: DDF-O: PVP paint applied to different substrates. (**c**) Flexible light-emitting film of MODPA: DDF-O: PVA.

**Table 1 molecules-29-04537-t001:** Electrochemical data for the compounds and frontier orbital energies (HOMO and LUMO).

Compounds	E_onset_ (V)	E_onset_ (V)	E g_op_ (eV)	E_HOMO_ ^a^ (eV)	E_LUMO_ ^b^ (eV)
MODPA	-	0.77	3.55	−5.70	−2.15
DDF-O	-	0.17	3.22	−5.10	−1.88
DDF-CHO	-	0.27	3.06	−5.20	−2.14
DDF-Br	-	0.37	3.15	−5.30	−2.15
DDF-TRC	−1.14	0.40	403	−5.33	−3.79

^a^ The HOMO energy was deduced from the oxidation onset potential from cyclic voltammetry data and calculated by the equation: E_HOMO_ = −E_onset_ − 4.93 (eV). ^b^ The LUMO energy was deduced from the reduction onset potential from cyclic voltammetry data and calculated by the equation: E_LUMO_= −E_onset_ − 4.93 (eV) or E_LUMO_= E_HOMO_ + E_gop_ [24,28,29,30], where E_onset_ and E_onset_ are the first oxidation and reduction half-wave potentials, respectively. The optical band gap (Eg_op_) was estimated from the wavelengths corresponding to intersection point between normalized absorption and fluorescence spectra, as shown Appendix A.

## Data Availability

The original contributions presented in the study are included in the article/Appendix A, further inquiries can be directed to the corresponding author.

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
