# Peer review of "Study on the Influence of Host–Guest Structure and Polymer Introduction on the Afterglow Properties of Doped Crystals"

_molecules, 2024, doi:10.3390/molecules29194537_

Round 1

Reviewer 1 Report

Comments and Suggestions for Authors

The paper presents an interesting study on the influence of host-guest structure and polymer introduction on the afterglow properties of doped organic crystals. The findings, such as the ability to tune the afterglow duration by modifying the guest molecule structure and further enhancing the afterglow by using a polymer matrix, are significant. The paper is well-organized, and the writing is clear. The introduction provides a good background on organic long-persistent luminescence materials and the importance of the research topic. The experimental, results, and discussion sections are structured logically, and the figures are of high quality and effectively support the text.

Overall, this is a well-designed and executed study that advances the understanding of organic long-persistent luminescence materials. The findings are significant, and the paper is well-written. I recommend the publication of this manuscript after addressing the minor suggestions mentioned below:

The authors could consider the following potential improvements:

1.      Line 259: “The FL emission attenuation curve further proves that charge transfer occurs between host and guest”. Please discuss it in detail.

2.      Explore any potential structure-dependent trends in the afterglow duration and intensity and provide a more comprehensive structure-property correlation.

3.      In the conclusion section, outline future research directions or strategies to further improve the afterglow performance of the materials.

4.      Discuss the possibility of further improving the afterglow properties, such as by combining the host-guest strategy with other approaches (e.g., molecular engineering, doping optimization).

Comments on the Quality of English Language

Minor editing of the English language is required.

Reviewer 2 Report

Comments and Suggestions for Authors

   The manuscript is well written in understandable English with only several grammatical or improper wording errors. However, it also contains some meritorious errors that need to be corrected to make the manuscript acceptable for publication.  The following items require correction:

1. The label 'MODPA" is missing in Fig.1a (in vicinity of 4.4'-dimethoxydiphenylamine structure).  Without that label, the acronym MODPA is undefined.

2. The vertical axis title of  Fig.1d should be: "Log(Intensity) / a.u.", because according to the figure title, it is a semi-logarithmic plot.

3. line 123: The temperature 160 oC is probably wrong (or was too low), because PMMA melts above 200 oC.

4. line 125: Polyethylene glycol (PEG) available commercially may have different molecular weights and properties (e.g., PEG 300, PEG 1000, PEG 3000, etc, where the number denotes its molecular weight).  Hence, the molecular weight of the PEG used by the authors should be specified somewhere to make the results reproducible.

5. Vertical axis titles are missing in Fig.2a and Fig.2b. 

6. There is a wrong unit of current density in Fig.2d, because at the current density of the order of 100 A/cm2, the sample tested would ignite in air and burn up (or evaporate) immediately.  This was probably mA/cm2.  Moreover, axes titles in Fig.2d insert are missing.  Also Fig.2d title is wrong, because the figure shows current density (J) versus voltage curves, (not current-voltage traces), while it is not clear what the authors mean by "trap density" (not shown in Fig.2).

7. Colors of particular plots specified in Fig.3a title are not consistent with the colors shown in that figure (i.e., the authors specified that Fig.3a shows absorption (yellow), fluorescence (blue) and phosphorescence (red) spectra, while the colors of those spectra seen in Fig.3a are: orange, cyan and magenta, respectively. Probably the authors plotted original spectra using RGB color coordinates, which were treated as CMY color coordinates during conversion to PDF file, that resulted in false colors. Consequently, the color names in Fig.3a legend should be appropriately adjusted to what is actually seen in that figure.

8. The emission lifetimes specified in Fig.3b legend are not consistent with the corresponding slopes of the plots shown in that figure.  For example, Fig.3b indicates that MODPA:DDF-CHO (orange data points) had the longest emission lifetime (as indicated by its lowest slope), while the lifetimes reported in the legend are highest for  MODPA:DDF-O.  Either the lifetimes are miscalculated or the figure legend is messed up.

9. The authors used glass (or PMMA) cuvettes for recording absorption spectra of the compounds studied (as indicated by the steep fall of  the absorbance and noisy baseline below 300 nm, in Figures 3a, S1 and S2).  Consequently, the absorbance plots should start from 300 nm (because in the case of glass cuvettes, the absorbance data below 300 nm are meaningless due to strong disturbance of the spectra by absorption of the cuvette material).  If the absorption spectra are supposed to reach below 300 nm, a quartz cuvette has to be used.

10. Vertical axis titles of Figures S1 and S2 (in Supplementary Information) are wrong, because absorbance is not an "intensity" (i.e., intensity is a parameter of emission spectra).  "Absorbance" is the correct word.  Moreover, the spectra shown in Fig.S2 are not normalized and the baseline of DDF-TRC is not compensated to the same level as that of the other spectra.

11. The absorption spectrum of MODPA, shown in Fig.S1 is misleading, because it was recorded in a glass (or PMMA) cuvette. Consequently, it is cut off at the short wavelength edge and the MODPA absorption maximum read out from Fig.S1 (299 nm) is not consistent with the absorption maximum reported in line 207 (283 nm).  So, it is not clear where the value 283 nm came from. Quartz cuvette is necessary for reliable determination of the MODPA long-wavelength absorption maximum.

12. lines 211-212:  The authors mention "nitro electron-withdrawing  capacity" of DDF-TRC, while there is no nitro group in the DDF-TRC structure.

13. The statement in lines 218-222 does not make sense, because the positions of fluorescence or phosphorescence peaks  have nothing in common with the absorption magnitude (called by the authors "strong absorption").

14. lines 234-235: The authors wrote that "the energy gaps are decreasing progressively for DDF-O, 234 DDF-CHO, DDF-Br, and DDF-TRC, while Fig.4 shows that the energy gap of DDF-Br (labeled incorrectly as DDF-4Br) is higher than that of DDF-CHO.  So, the energy gaps decrease in different order than that specified.

15. line 237: It is not clear what the authors mean by "the physical process of host-guest spectroscopy."

16. The column E1/2-/2- in Table 1 may be deleted, because it contains no data. Moreover, the limited data reported in columns E1/20/- and E1/22+/+ are neither discussed nor used for any calculations, so these columns are unnecessary too.

17. line 241: Reference to the source of the equation: "EHOMO = -Eonset – 4.93", has to be added, because the magnitude of the constant "-4.93 eV" depends on the type of reference electrode used for determination of  Eonset. The authors used A/Ag+ reference, so it is not clear whether the constant "-4.93 eV" is specific for the Ag/Ag+ reference, or perhaps to SCE or NHE references. The large discrepancies between ELUMO–EHOMO energy gaps calculated  from the data in Table 1 and the corresponding energy gaps reported in Fig.4 suggest that some of the calculations are incorrect.

18. The cyclic voltammetry data (Fig.S3) indicate that the data contained in column E1/2+/0 of Table 1 are not standard oxidation potentials of the compounds studied (vs. Ag/Ag+ reference) as the E1/2 symbol would imply.  These are onset oxidation potentials (Eonset), subsequently used for calculation of EHOMO. Hence, the header E1/2+/0 needs to be changed to Eonset.  Moreover,  the oxidation potential of DDF-O is missing.

19. The wavelengths reported in Table 1 under the header lmax(abs) are shifted towards longer wavelengths compared to the corresponding absorption maxima reported in line 207.  This means that "lmax(abs)" data in Table 1 represent absorption onset positions, (i.e, not the position of the absorption maxima, as the header would imply), which are subsequently used for estimation of the HOMO-LUMO energy gap (labeled as Egop ).  Consequently, the header lmax(abs) should be changed to lonset(abs) to avoid misinterpretation of the data.  Moreover, estimation of the optical bandgap (Egop) would be more accurate if the authors used the wavelengths corresponding to intersection point between normalized absorption and fluorescence spectra, instead of some long-wavelength absorption onset position.  The onset is used only when the compound does not fluoresce.

20. The statement in lines 262-265 does not make sense, because the charge transfer processes between two molecules and the intersystem transitions from singlet to triplet within the same excited molecule are not radiative transitions.  The emission of photons with longer emission lifetime may accompany back electron transfer within the MODPA/DDF-O charge-separated (CS) states, while the shorter lifetime may be attributed to the emission from singlet excited states of  DDF-O alone.

21. lines 300-301: The statement end: "...which indicates that this kind of PMMA material has good doping properties." is unclear, because it is not clear what the authors mean by "doping properties".

Comments on the Quality of English Language

The following replacements or deletions will improve some of improper wording or grammar:

- line 42: delete: synthesis (because metals are not "synthesized", while the synthesis cost of LPL materials based on iridium or platinum complexes is negligible compared to the price of these metals.)

- line 80: delete: physical (redundant)

- line 111: there is: 1 mol% MODPA:DDF-O -> there should be: 1mol% DDF-O/99mol% MODPA (because "1 mol% MODPA:DDF-O" would suggest that MODPA was a minor component)

- line 112: dispersed -> prepared (because a powder could be "dispersed", not the solution)

- line 135: stirring -> stirred

- line 142: Ocean optical -> Ocean Optics (because this is the manufacturer name)

- lines 145/146: fluorospectrophotometer -> fluorescence spectrometer

- line 184: lamellar -> lamellar structure ; Mobility -> Charge carrier mobility

- line 196 and Fig.S1 title: measuring -> measured

- lines 194/195: delete: emission (because absorption specified at the beginning of the spectra list is not an "emission spectrum")

- Fig.4: delete: and the corresponding energies are provided in the parentheses (because there are no parenthesis in that figure), and add the title: "Energy" to the energy axis.

- Table 1: ELMOb -> ELUMO ; EHOMOa -> EHOMO

- line 263: DDF2o -> DDF-O

- line 286: copolymers -> polymers (because PMMA, PEG and PVP are not "copolymers")

- line 286: substrates -> hosts

- line 287: doping -> dopants

- line 292: crystals -> materials (because the polymers used do not form crystals)

- line 297: base -> host

- line 330: apply -> applied

- Fig.S3: Povential -> Potential

Reviewer 3 Report

Comments and Suggestions for Authors

In this manuscript host-guest materials were prepared with a view to afterglow effects, also in combination with common polymers. Thus, variable afterglow times on the time scale of several seconds were achieved. The materials were investigated with various methods. The effect of afterglowing was clearly demonstrated. I recommend publication of the manuscript after revision as indicated below.

p. 1, Introduction, line 1 - 3 (total line 31 - 33), and p. 1, Introduction, line 9 - 13 (total line 38 - 43): The statements about toxicity are problematic. Certainly, organic materials are not environmentally friendly in general, whereas the toxicity of platinum cannot be so drastic as it is a highly estimated metal in jewelry.

p. 1, Introduction, line 13 – 14 (total line 43 – 44) / p. 2, Introduction, line 1 (total line 45): The general statement that organic materials are simple to synthesize does not hold. In fact there are organic materials that are very difficult to synthesize. Also, organic materials are not in general of low cost (there are expensive organic materials), and molecular weight regulation is not easy for all organic materials, and not all organic materials show low toxicity to organisms.

p. 3, section 2.1 Materials (total lines 105 – 109): The purification procedure of MODPA is not described. Further it is unclear what is understood of the “standard methods” that were used to purify the four compounds. The purifications should be described in detail. Also it should be carefully described how the purity of ≥99.9% was measured.

p. 5, line 6 – 8 (total line 180 – 182), and Figure 2b: The authors say that the transitions in the DSC traces in Figure 2b are due to melting. However, this should be proven by DSC traces upon cooling. In case of melting the transition to the solid state would be observed typically at somewhat lower temperature than the temperature upon melting.

p. 5, line 9 – 10 (total line 183 – 184), and Figure 2c: According to the authors, fluorescence microscope images show that the doped crystals have lamellar structure. However, this is not evident from the images (Figure 2c). These images only show bright areas without any insight in the structures. In order to investigate the structure, XRD diffraction should be performed, which would give evidence for lamellar structures if present.

 p. 9, line 2 – 3 (total line 196 – 197), and 5th / 4th line before the bottom (total line 317 – 318): The polymers poly(methyl methacrylate), poly(ethylene glycol) and poly(vinyl alcohol) are not mentioned in the section 2.1. Materials. The source of these polymers as well as their average molar masses should be indicated (number average molar mass or weight average molar mass). In addition, the degree of hydrolysis of poly(vinyl alcohol) should be indicated as poly(vinyl alcohol) is commonly prepared from hydrolysis of poly(vinyl acetate) (the hypothetical monomer vinyl alcohol essentially does not exist, it is the tautomeric form of ethanal and is present only in a negligible fraction).

Comments on the Quality of English Language

Minor editing of English language would be favorable.

Round 2

Reviewer 2 Report

Comments and Suggestions for Authors

I have no further comments.

Reviewer 3 Report

Comments and Suggestions for Authors

The manuscript has been revised adequately, although the DSC curves upon cooling are still missing in Figure 2b. However, as Figure 2a shows a mass loss well above the transitions evident in Figure 2b the attribution of these transitions to melting processes seems to have a probability which is not entirely unreasonable.